# Relevance of Dietary Supplement Use in Gastrointestinal-Cancer-Associated Cachexia

**DOI:** 10.3390/nu15153391

**Published:** 2023-07-30

**Authors:** Saunjoo L. Yoon, Oliver Grundmann

**Affiliations:** 1Department of Biobehavioral Nursing Science, College of Nursing, University of Florida, Gainesville, FL 32610, USA; yoon@ufl.edu; 2Department of Medicinal Chemistry, College of Pharmacy, University of Florida, Gainesville, FL 32611, USA

**Keywords:** cancer cachexia, gastrointestinal cancer, dietary supplements, nutritional supplements

## Abstract

Cancer cachexia is a multi-organ syndrome with unintentional weight loss, sarcopenia, and systemic inflammation. Gastrointestinal (GI) cancer patients are more susceptible to cachexia development due to impaired nutrient absorption and digestion. Given the widespread availability and relatively low cost of dietary supplements, we examined the evidence and effects of fish oil (omega-3 fatty acids), melatonin, probiotics, and green tea for managing symptoms of GI cancer cachexia. A literature review of four specific supplements was conducted using PubMed, Google Scholar, and CINAHL without a date restriction. Of 4621 available literature references, 26 articles were eligible for review. Fish oil decreased C-reactive protein and maintained CD4+ cell count, while melatonin indicated inconsistent findings on managing cachexia, but was well-tolerated. Probiotics decreased serum pro-inflammatory biomarkers and increased the tolerability of chemotherapy by reducing side effects. Green tea preparations and extracts showed a decreased risk of developing various cancers and did not impact tumor growth, survival, or adverse effects. Among these four supplements, probiotics are most promising for further research in preventing systemic inflammation and maintaining adequate absorption of nutrients to prevent the progression of cancer cachexia. Supplements may benefit treatment outcomes in cancer cachexia without side effects while supporting nutritional and therapeutic needs.

## 1. Introduction

Gastrointestinal (GI) cancers are among the most common cancers diagnosed (17.8%) and with the highest mortality rate (28.2%) among all cancers [1]. Involuntary weight loss is common among GI cancers [2], with advanced GI cancer patients at an increased risk of developing cancer cachexia. Cancer cachexia is commonly defined as ≥5% weight loss within a 6-month period [3]. The overall incidence of developing cancer cachexia in GI cancer patients is between 40% and 80% [4]. The risk of cancer cachexia in GI cancer patients may be as high as 90% in pancreatic cancer and as low as 15% in prostate cancer patients [5,6]. The prevalence of cachexia is inversely correlated with 5-year survival, indicating a tumor’s site-specific contribution [7]. Patients with lower body mass index (BMI) are at a higher risk of impacted quality of life and reduced survival when developing cancer cachexia compared to patients with a normal or high BMI [8].

In recent years, the early detection of cancer cachexia in patients has been of interest to improve survival and quality of life. Multiple factors contribute to a delayed diagnosis of GI cancer. Sudden and unexpected weight loss may be the first sign preceding a GI cancer diagnosis. At this point, the patient is more likely to be already pre-cachectic or in a cachectic condition and be diagnosed with an advanced GI cancer stage. Screening for colorectal and other GI cancers is based on national guidelines, with those for the United States (US) advising regular screening starting at 50 for both men and women [9]. Although most colorectal cancers are more likely to occur in ages 50 and older, the incidence among those younger than 50 accounts for about 0.5% [10]. Environmental factors contributing to the development of GI cancers, such as alcohol and tobacco use, obesity, and a diet high in fat and low in fiber, may further increase the individual risk [11].

Cancer cachexia is a multi-organ syndrome with unintentional weight loss, sarcopenia, and inflammatory processes as its hallmarks. The breakdown of lean muscle mass is often observed independent of cachexia with advanced age. Furthermore, it limits the quality of life and physical functioning and increases mortality in patients with advanced cancer. Knowing inflammatory processes accompany weight and lean muscle loss, it has been proposed that dysregulation of pro- and anti-inflammatory proteins in conjunction with metabolic hormones are causative in the development of cancer cachexia because of the tumor microenvironment [12].

Another contributing factor especially prevalent among patients with GI cancer is malnutrition, a lack of sufficient caloric intake, and/or insufficient provision of macro- and micronutrients. In the case of cancer cachexia, malnutrition as a precursor is linked to low protein intake, which is of particular concern given the already accelerated and potentially irreversible loss of lean muscle mass. Chemotherapy-related lack of appetite, nausea or vomiting, and GI upset further increase the risk of malnutrition accompanying cancer cachexia.

Because of malnutrition, GI cancer patients require an in-depth nutritional needs assessment to evaluate the risk of developing cachexia [13]. Indeed, pancreatic cancer patients with *any nutritional risk* have significantly lower survival than those without [14]. Specific populations, including patients with diabetes and obesity, are at an increased risk of developing cancer-associated cachexia, given their pre-existing conditions [15]. Many disease conditions are associated with increased systemic inflammation that impacts body composition. By evaluating body composition by anthropometric means and systemic inflammation biomarkers (e.g., albumin, white blood cell, neutrophile, lymphocyte, and platelet counts), the risk of cancer cachexia and need for nutritional intervention have been established [16]. A recent meta-analysis of studies evaluating the impact of nutritional supplementation on pro-inflammatory biomarkers and length of hospital stay in colorectal cancer patients demonstrated that glutamine was superior in reducing tumor-necrosis factor α (TNF-α) and shortening hospital stay, while probiotics reduced the incidence of pneumonia [17]. One intervention that has shown to be effective in reducing postoperative complications, including malnutrition and cachexia, is preoperative immunonutrition (especially those containing glutamine, arginine, and omega-3 fatty acids) [18,19].

Cancer cachexia is characterized by the co-occurrence of decreased energy intake and increased energy expenditure, leading to a negative energy balance. The primary contributors to reduced energy intake are loss of appetite (anorexia), dysphagia, pain, fatigue, and depression or anxiety [20]. Increased energy expenditure is caused by tumor metabolism, systemic inflammation, and decreased energetic efficiency due to metabolic dysregulation [20]. The negative energy balance, in turn, leads to insulin resistance and oxidative stress, which further facilitate inflammation. The primary pro-inflammatory cytokines related to the development of cancer cachexia are TNF-α, interleukin (IL)-1 beta, IL-6, epidermal growth factor (EGF), transforming growth factor (TGF)-β, and platelet-derived growth factor (PDGF) [21]. The pro-inflammatory mediators are released by the tumor microenvironment and systemically. It leads to reduced muscle protein synthesis by downregulating the mammalian Target of Rapamycin (mTOR) and increased muscle degradation by upregulating atrogin-1 and Muscle Ring-Finger Protein-1 (MuRF-1).

Current treatment of cancer cachexia remains limited to short-term prevention of progressive muscle degradation and increasing protein synthesis. As such, non-steroidal anti-inflammatory drugs (NSAIDs) are used to reduce the release of pro-inflammatory mediators and cancer-associated pain, often leading to anorexia. Steroids, in particular corticosteroids, to reduce systemic inflammation and megestrol acetate for appetite stimulation often provide short-term improvement. Similarly, anamorelin hydrochloride, an orally administered drug with ghrelin-like effects, can be used to stimulate appetite [22]. Other potential treatment options include physical exercise, targeted acupuncture, nutrition therapy, as well as enteral and parenteral nutrition.

With more than 60% of adults in the US using at least one dietary or herbal supplement [23], their use to benefit GI cancer patients and especially aid in preventing or treating GI cancer cachexia is promising, given their widespread availability and relatively low cost. 

While nutritional supplementation is often incorporated into pre- and post-surgical treatment of cancer patients as the standard of care to reduce the risk of post-surgical weight loss and cachexia [24], dietary supplements cannot be used with a clinical indication or considered clinical intervention based on the US Dietary Supplement Health and Education Act (DSHEA), which prevents them from being labeled with an indication [25]. Instead, dietary supplements are regulated differently from nutritional supplements without a disease indication and are often regarded suspiciously by healthcare professionals and patients alike. Nonetheless, patients often used dietary supplements in addition to the standard of care with or without reporting to their healthcare providers [26].

In this narrative review, we examine the evidence of fish oil (omega-3 fatty acids and, in particular, eicosapentaenoic and docosahexaenoic acid), melatonin, probiotics, and green tea in preventing and managing GI cancer-related symptoms and developing cancer cachexia, which presented the most robust preclinical and clinical research with cachexia in GI cancer.

## 2. Methods

### 2.1. Search Strategy

The databases searched were PubMed, Google Scholar, and CINAHL, without a date restriction. The search terms included combinations of the phrases “gastrointestinal cancer” or “digestive cancer” AND “cachexia” AND either “supplements”, “fish oil”, “melatonin”, “probiotics”, or “green tea”. A total of 4621 articles were identified. Of those, 130 publications were from PubMed, 4472 from Google Scholar, and 19 from CINAHL (Table 1). The literature search followed Preferred Reporting Items for Systematic Reviews and Meta-Analysis (PRIMSA) guidelines [27]. We excluded 635 duplicates, 217 published in a language other than English, 3630 that were not related to gastrointestinal or digestive cancer and cancer cachexia, and 113 unrelated review articles. A total of 26 articles, including original preclinical and clinical research articles and reviews, were part of this review (Figure 1).

### 2.2. Study Inclusion and Exclusion Criteria

Inclusion criteria were (1) the studies focusing on cachexia related to GI or digestive cancers, (2) observational, cross-sectional, randomized controlled trials, and (3) preclinical or clinical studies. Exclusion criteria were systematic reviews, narrative reviews, conference proceedings, non-English articles, articles not related to cancer-associated cachexia, and supplements used unrelated to GI or digestive cancers.

## 3. Results

The available clinical evidence supporting the use of dietary or herbal supplements in GI cancer patients with cachexia is limited to small interventional or observational studies. In addition, some supplements were used as part of a nutritional intervention which complicates assessment as a direct factor in reducing or reversing the progression of cancer cachexia. Of those 26 articles included in this review, fish oil was the most commonly studied (6 clinical articles and 6 preclinical), followed by 15 probiotic-focused studies (3 clinical and 12 preclinical, 10 green tea studies (no clinical study), and 9 studies on melatonin (4 clinical and 5 preclinical).

### 3.1. Fish Oil

Fish oil is rich in omega-3 fatty acids, which aid in energy metabolism and utilizing fatty acids high in energy density. It has been shown to reduce levels of pro-inflammatory mediators because fish oil contains high amounts of omega-3 fatty acids, eicosapentaenoic acid (EPA), docosahexaenoic acid (DHA), and other polyunsaturated fatty acids (PUFAs) [28]. In a study of GI cancer patients undergoing surgical intervention, adding fish oil to arginine improved post-surgical outcomes and shortened recovery compared to no nutritional support in 305 patients [29]. Interestingly, this study found no difference in patients that were given the supplements only prior to surgery compared to those given the supplements both prior to and following surgery. This may indicate a preventive effect of fish oil on systemic inflammation. Although not statistically significant, patients given fish oil both pre- and post-surgery lost less weight compared to no supplementation or pre-surgery supplementation only. In a review on the use and effect of omega-3 fatty acids and fish oil preparation in cancer cachexia, a number of included studies reported an increase in body weight with fish oil supplementation; however, inflammatory parameters were either not impacted or the change was not uniform between different studies [30]. Contrary to this review, a clinical study in patients with breast cancer indicated lower plasma high-sensitivity C-reactive protein and maintaining CD4+ T lymphocytes in the experimental group with fish oil enriched in EPA and DHA supplementation for 30 days compared to a control group whose CD4+ lymphocytes significantly decreased [31]. However, other pro-inflammatory markers remained unchanged between the groups, supporting a differential impact of fish oil supplementation on biological markers and patient outcomes (Table 2).

A placebo-controlled study in 128 GI cancer patients with cachexia indicated that fish oil-enriched nutritional support leads to lower C-reactive protein blood levels while increasing skeletal and lean muscle mass compared to the placebo group over 6 months of treatment [32]. The group receiving fish oil also showed improved chemotherapy tolerance compared to the control group, indicating a direct benefit of therapeutic outcome. The findings of this study indicated that overall survival was not significantly different between the fish oil supplement group and the placebo group [32]. However, a subgroup of patients with the modified Glasgow Prognostic Score (mGPS) of 1 or 2 benefited the most from nutritional supplementation of fish oil by significantly prolonging survival time compared to the same subgroup without the supplementation [32].

In a small clinical study of 33 patients with pancreatic cancer who had developed cachexia, fish oil and marine phospholipids were compared to examine the improvement of quality of life and weight stability [33]. Both groups received an equal amount of 0.3 g omega-3 fatty acids over a course of 6 weeks. Both groups presented with similar outcomes, indicating the benefit of omega-3 fatty acids in preventing weight loss and stabilizing lean muscle mass. In both groups, serum C-reactive protein levels were reduced by more than 50%, while high-density lipoprotein increased by 22% only in the fish oil group. Furthermore, only the fish oil supplement group showed a positive correlation with improved global quality of life as the concentration of EPA increased over the study period. 

A triple-blind randomized clinical trial compared fish oil that contained 1.0 g EPA and 0.5 g DHA in 3.6 g of total supplement to olive oil without either ingredient or polyunsaturated fatty acids in 45 cachectic patients with colorectal or gastric cancer over nine weeks [34]. While the global quality of life decreased for both groups, fatigue and nausea/vomiting were significantly worse in the olive oil group compared to the fish oil group for both types of GI cancers. Appetite loss was significantly higher in the olive oil group, although it also increased in the fish oil group, albeit not clinically significant. The authors point to limitations of the study given that olive oil may also contain anti-inflammatory compounds such as oleic acid that make a definitive determination of the beneficial effects of fish oil difficult. 

A clinical study of 125 advanced cancer patients with cachexia compared five study arms (medroxyprogesterone vs. EPA/DHA vs. L-carnitine vs. thalidomide vs. combination of all). Among these, the EPA/DHA arm did perform worse compared to L-carnitine, thalidomide, or combination therapy in regard to lean body mass, resting energy expenditure, and fatigue [35]. The authors concluded that combination therapy provided the greatest benefit to patients. Of note, thalidomide is not approved for the treatment of cancer cachexia, and the current evidence remains unclear as to its benefit-to-risk ratio, given its genotoxicity [36].

A randomized pilot study compared (1) the effects of enriched fish oil (30 mL/d), containing 4.9 g of EPA and 3.2 g of DHA to melatonin (18 mg/d) supplementation for four weeks, and (2) followed by combined fish oil and melatonin to all patients for the next four weeks in a small sample of 24 GI cancer patients [37]. The fish oil group (5 out of 13 patients: 38%) showed a numerically higher percentage of patients maintaining their body weight than those in the melatonin group (3 out of 11 patients: 27%), while 63% of patients after combination therapy had stable or gained weight. However, it is not clear whether a total of 8 weeks of treatment or a combination of the two results in more positive outcomes (Table 2).

According to a review article on the use of fish oil in patients with advanced cancer, the evidence available as of 2012 does not clearly indicate a benefit of fish oil [38]. The authors stated that fish oil did present with a low adverse effect risk that was usually tolerable by patients. In rare instances, fish oil was discontinued if it resulted in nausea, constipation, or reduced appetite. In 2018, a systemic review of the use of omega-3 products indicated a favorable outcome for patients who took 2 g/day of EPA, resulting in lower systemic inflammation and weight stabilization or gain [39]. Although the data were not conclusive, the authors stated that future studies should consider more homogenous populations by specific cancer types and stages to distinguish beneficial effects.

The use of fish oil may benefit patients with GI cancers who are either pre-cachectic or developed cancer cachexia and abnormal CPR or albumin levels (mGPS 1 or 2), at least regarding weight maintenance with minimal potential side effects. Because of the different doses of EPA/DHA used and varying trial period lengths, it is difficult to definitively conclude what composition of fish oil over what period of time will benefit patients. At a minimum, a fish oil preparation should contain EPA and DHA in a ratio of 2:1 and a minimum dose of 2 g/day over at least a six-week period to establish a beneficial effect.

### 3.2. Melatonin

Melatonin is a hormone released from the pineal gland that serves many functions, regulating the wake–sleep cycle as well as neuroimmunomodulation [40]. Because of its cyclic release, melatonin fundamentally impacts cortisol release, which in turn relates to immune system regulation. Thus, melatonin release and supplementation may correlate with decreased immune system suppression or inflammation. In a clinical trial with 86 patients with various types of solid tumors, patients with cancer cachexia benefited from additional melatonin supplementation compared to supportive care alone for three months to reduce TNF-α serum levels and weight loss [41].

Because of its critical role in the circadian rhythm, melatonin also impacts the autonomic nervous system to counteract sympathetic outflow at night and reduce norepinephrine and epinephrine blood levels [42]. It leads to reducing the release of endogenous cortisol from the adrenal cortex to prevent systemic stress signaling [43]. Higher endogenous concentrations of norepinephrine and epinephrine have also been reported in heart failure patients with cardiac cachexia [44].

A small clinical study in nine pediatric patients with solid tumors indicated that melatonin in doses up to 10 mg or 0.3 mg/kg over 8 weeks was well tolerated and resulted in weight gain in seven out of nine patients [45].

Melatonin has been shown to regulate food intake by binding to melatonin receptors in the GI tract and is excreted directly into the intestine during the daytime [46]. A study comparing supplementation with melatonin 20 mg/day to placebo in patients with cancer cachexia was stopped after 28 days, indicating a lack of efficacy with no changes in body weight, sleep, or appetite compared to the control group [47]. However, melatonin did not lead to any toxicity in patients (Table 2).

While high-dose melatonin supplementation has not consistently been shown to benefit patients with GI cancer cachexia, it may improve the effectiveness of concomitant chemotherapy and reduce epidermal growth factor receptor activation to prevent metastasis of localized cancer [46,48]. Given its good tolerability with few adverse effects, melatonin may be trialed in patients with GI cancer cachexia in combination with other interventions, such as nutritional counseling and pharmacotherapy. Because of the dearth of clinical trials with melatonin, no recommendation on its use can be made at this time in the prevention or treatment of cancer cachexia in GI cancers. It does appear to decrease weight loss and reduce inflammation in doses of 10 mg/day or more over at least an eight-week period.

### 3.3. Probiotics

Probiotics are live microorganisms used as food items (e.g., probiotic yogurt) and as dietary supplements to benefit overall well-being, mitigate intestinal inflammation, and maintain or improve gastrointestinal health [49]. Specifically for cancer, few studies have evaluated the potential benefit of probiotic supplementation. Probiotics may restore intestinal barrier function, which is often weakened during chemotherapy due to systemic and localized inflammation [50]. Furthermore, patients with GI cancers may have to undergo surgery which causes dysbiosis of an already weakened gut microbiome. In such cases, the European Society for Clinical Nutrition and Metabolism recommends the use of multi-strain probiotics in combination with prebiotics (most commonly inulin as a nutrition source for bacteria) to reduce the risk of infections and intestinal inflammation [51].

The generation by intestinal microbiota of branched-chain amino acids that contribute to muscle anabolism and short-chain fatty acids that provide for anti-inflammatory activity and serve as energy sources for enterocytes has been found to be deficient in the gut of patients with cancer cachexia [52]. The composition of the gut microbiome appears to shift in cachectic individuals, as shown in a number of preclinical animal studies [53]. Especially, a decrease in *Clostridiales* and *Lactobacillus* and an increase in *Bacteroidetes* and *Enterobacteriales* has been observed in cachectic animals compared to healthy controls. In humans, the administration of *Lactobacillus* genus appears to have benefited patients with cancer in reducing or treating cachexia [52]. Animal models confirmed that *Lactobacillus* administration reduced muscle atrophy markers and the production of systemic pro-inflammatory cytokines [54].

The *Lactobacillus rhamnosus* probiotic strain increased the effectiveness of the chemotherapeutic agent, capecitabine, in a mouse model of colon cancer if administered prior to inoculation of CT-26 cells [55]. The study indicated that the probiotic not only increased the effectiveness of chemotherapy resulting in reduced tumor size, but also decreased chemotherapy-related adverse effects by preventing reduction in white blood cells, inducing apoptosis through increased caspase-3 levels, and reducing systemic inflammation indicated by decreased IL-6 levels [55]. A randomized, single-blind, placebo-controlled prospective study of 100 colorectal cancer patients showed that those who were given probiotics following surgery indicated a lower gastrointestinal dysbiosis, which improved quality of life and reduced chemotherapy-related malnutrition [56].

Probiotics digest certain sugars to produce short-chain fatty acids that serve as energy sources for enterocytes, thus restoring intestinal function. The probiotic supplementation may benefit patients receiving adjunct antibiotic therapy to reduce GI upset and prevent further weakening of the immune system [57,58]. This has also been confirmed in colorectal cancer patients who received a probiotic for six months following surgery compared to a placebo [59]. Several serum pro-inflammatory biomarkers (TNF-α, IL-6, IL-10, IL-12, IL-17A, IL-17C, and IL-22) were significantly reduced in the probiotic group compared to the placebo group after six months. Probiotics as part of enteral or parenteral nutritional support appear to improve the quality of life in patients with GI cancer cachexia, although this appears to be limited to patients with good functional status [60].

A randomized clinical study evaluated the use of the probiotic strain *Lactobacillus rhamnosus* in 150 colorectal cancer patients receiving 5-fluorouracil chemotherapy compared to a placebo [61]. The probiotic group had less severe diarrhea, reported less severe abdominal pain and discomfort, needed less hospital care, and had fewer chemotherapy dose reductions due to gastrointestinal toxicity. A rodent study also indicated a reduction in intestinal irinotecan toxicity with supplementation of a probiotic bacterial strain mix, referred to as VSL#3 [62]. VSL#3 reduced weight loss and prevented or lessened diarrhea while increasing crypt proliferation to restore intestinal barrier function (Table 2).

Intestinal dysbiosis has also been associated with an increased risk for the development of cancer cachexia [63]. The contribution of the gut microbiome in preventing systemic inflammation and maintaining adequate absorption of nutrients is a critical factor in preventing the development and progression of cancer cachexia. Systemic inflammation, both because of the malignancy and chemotherapy, causes muscle wasting resulting in a negative quality of life. At this point, the use of probiotic strains in patients with GI cancer is beneficial given the alleviation of chemotherapy-associated GI adverse effects. This in turn may lead to maintenance of appetite and subsequent prevention of weight loss. Probiotics need to be supplemented as soon as possible and taken continuously throughout the treatment phase.

### 3.4. Green Tea

Initial evidence from animal studies [63,64,65] and one clinical study [66] indicate the benefit of green tea supplementation in reducing the risk of cachexia development in cancer patients. Animals given the green tea compound epigallocatechin-3-gallate did not develop skeletal muscle atrophy in a tumor-bearing model. In fact, the compound could even reverse muscle loss if given in higher doses to animals that had already developed cachexia [64]. An in vitro study concluded that green tea catechins and omega-3 fatty acids such as EPA and DHA downregulate toll-like receptor 4 signaling, induced in cancers to facilitate cachexia [65]. Because of the antioxidant and prooxidant activity of polyphenols present in green tea and turmeric, it has been hypothesized that the supplementation with such plant extracts may exacerbate an existing malignancy. A study of colon- and lung-tumor-bearing mice administered either fish oil, turmeric, green tea, or a combination of all did not impact tumor growth, survival, or result in any adverse effects [67]. However, this study did not report the benefit of such supplementation if animals already had developed malignancy and subsequent weight loss. Therefore, it is unclear whether supplementation with green tea or epigallocatechin-3-gallate may benefit patients with existing GI cancer cachexia.

Skeletal muscle maintenance is primarily based on the balanced process between protein production and proteolysis. This balance is skewed towards protein degradation in cachexia, a cascade of cellular mechanisms involving the ubiquitin-proteasome pathway [68]. Chief among them is atrogin-1, a F-BOX proteolytic enzyme, which is expressed as a consequence of increased TNF-α and reactive oxygen species levels to promote protein degradation. Several F-BOX proteins have been linked to cancer development and progression, as they are linked to glucocorticoid receptor-induced muscle wasting [69]. Green tea polyphenols, most prominently epigallocatechin-3-gallate, have been shown to downregulate genes encoding for atrogin-1 and other F-BOX proteins to prevent or reduce the progression of cachexia in vitro and in vivo [64,70].

According to the Cochrane review, the overall incidence of cancers with the consumption of high amounts of green tea as part of the diet indicates that green tea may lower the risk of some cancers, such as prostate cancer [relative risk (RR): 0.73, 95% confidence interval (CI): 0.56–0.94], oral cancer (RR: 0.71, CI: 0.62–0.82), any gastrointestinal cancer (RR: 0.78, CI: 0.59–1.02), colorectal cancer (RR: 0.84, CI: 0.74–0.96), or colon cancer (RR: 0.89, CI: 0.90–0.98) [66]. Another publication reviewed studies on different types of cancer and how regular green tea consumption may reduce that risk [71]. These reviews show that green tea consumption does reduce the risk of several cancers, despite only being marginally significant for some GI cancers. The evidence for green tea benefiting patients with GI cancer cachexia is not clear at this point.

**Table 2 nutrients-15-03391-t002:** Summary of included clinical studies on cancer type, study design, sample size, intervention (fish oil, melatonin, and probiotics), and primary outcomes.

Authors [Ref]	Cancer(s) and Treatment	Study Design (Intervention Implementation)	Sample Size	Supplement(s) with Doses	Outcome(s)
Moskovitz et al. [29]	GI	RCT, conventional (no supplementation), preoperative (supplementation prior to surgery), and perioperative (supplementation prior to and following surgery)	305 (102, 102, 101)	Oral Impact^®^ Immunonutrition (arginine, fish oil, and nucleotides)	Lower risk of infections and shorter hospital stay for both preoperative and perioperative supplementation
Shirai et al. [32]	GI and chemotherapy	RCT, no fish oil supplementation vs. fish oil supplementation (during chemotherapy)	128 (91, 37)	Fish-oil enriched supplementation containing 1.1 g EPA, 0.5 g DHA, 16 g protein (Prosure^®^)	26/37 fish oil patients and 53/91 non-fish oil patients completed chemotherapy, with increased skeletal and lean muscle mass in fish oil patients
Werner et al. [33]	Pancreatic and chemotherapy, radiation therapy, or supportive care	RCT, fish oil fatty acids vs. marine phospholipids (in any stage of the treatment)	33 (18, 15)	Fish oil (7% EPA, 13% DHA), marine phospholipids (8% EPA, 12% DHA)	Reduction in C-reactive protein in both groups, lower thrombocyte and LDL/HDL ratio in fish oil group, higher HDL in fish oil group, and quality of life correlates with increase in blood EPA levels
Mocellin et al. [34]	Lower GI and chemotherapy	RCT, olive oil control vs. fish oil capsules (started on the first day of chemotherapy and for the next 9 weeks)	45 (23, 22)	Olive oil (no EPA, DHA, or PUFAs) vs. fish oil (28% EPA, 14% DHA, 42% PUFAs)	Non-significant increases in body weight, fat free mass, and body water in the fish oil group
Mantovani et al. [35]	All cancers and advanced stage	RCT, medroxyprogesterone/megestrol vs. EPA vs. L-carnitine vs. Thalidomide vs. all combined (during antineoplastic chemotherapy or hormone therapy)	125 (21, 25, 24, 20, 20)	Medroxyprogesterone (500 mg)/megestrol (320 mg), EPA (2 g, with DHA), L-carnitine (4 g), thalidomide (200 mg)	Combination treatment of all interventions increased body weight, appetite, resting energy expenditure, and fatigue symptoms vs. each treatment alone; EPA alone was not effective
Mantovani et al. [72]	All cancers	RCT, medroxyprogesterone/megestrol vs. EPA vs. L-carnitine vs. thalidomide vs. all combined (during antineoplastic chemotherapy or hormone therapy with palliative intent or supportive care)	332 (44, 25, 88, 87, 88)	Medroxyprogesterone (500 mg)/megestrol (320 mg), EPA (2 g, with DHA), L-carnitine (4 g), thalidomide (200 mg)	Combination treatment of all interventions increased body weight, appetite, resting energy expenditure, and fatigue symptoms vs. each treatment alone; EPA alone was not effective, but sample size analysis did not meet goal of enrollment
Persson et al. [37]	Advanced GI	Non-blinded randomized study, fish oil vs. melatonin for 4 weeks followed by combination for 4 weeks (ongoing chemotherapy at least 2 courses)	24 (11, 13, 24)	Fish oil (4.9 g EPA, 3.2 g DHA), melatonin (18 mg)	Combination of fish oil and melatonin were superior to increase weight and improve QoL
Lissoni et al. [41]	Solid tumors and supportive care	RCT, standard treatment vs. standard treatment plus melatonin	86 (41, 45)	Standard treatment (opioids and steroids), melatonin (20 mg/d for 3 months)	Melatonin in addition to standard treatment did result in weight stabilization and reduction in TNF-α blood concentrations
Johnston et al. [45]	Solid tumors and chemotherapy	Phase 1 dose-escalation study in pediatric patients, melatonin for 8 weeks	9 (3 + 3 study design)	Melatonin in doses of 5, 7.5, and 10 mg	Average weight gain of 3.4% with melatonin use independent of dose while undergoing active chemotherapy cycles
Del Fabbro et al. [47]	Advanced lung or GI cancer (regardless of receiving treatment)	RCT, control vs. melatonin	48 (25, 23)	Melatonin (20 mg) vs. matched placebo for 4 weeks	Melatonin did not improve body weight, appetite, or QoL compared to placebo
Huang et al. [56]	Colorectal and chemotherapy	RCT, control vs. probiotic (6 weeks including 2 weeks of chemotherapy: from the third postoperative day to the last day of the first chemotherapy course)	100 (50, 50)	Probiotic (*B. infants*, *L. acidophilus*, *E. faecalis*, and *B. cereus*) vs. matched placebo for 6 weeks	Reduced chemotherapy-induced GI symptoms in probiotic group, restoration of disturbed gut microbiome
Zaharuddin et al. [59]	Colorectal and 4 weeks post-surgery	RCT, control vs. probiotic (starting twice daily for six months 4 weeks after colorectal resection)	55 (25, 30)	Probiotic (*L. acidophilus*, *L. lactis*, *L. casei*, *B. longum*, *B. bifidum*, and *B. infantis*) vs. matched placebo for 6 months	Significant reduction in pro-inflammatory cytokines in the probiotic group post-intervention without changes in interferon-γ
Österlund et al. [61]	Colorectal and post-surgery with no metastases receiving adjuvant chemotherapy or radiation therapy)	RCT, fiber vs. probiotic (during the 24 weeks of adjuvant cancer therapy)	150 (52, 98)	Probiotic (*L. rhamnosus*) vs. fiber (guar gum) during chemotherapy regimen	The probiotic group presented with significant less diarrhea but otherwise did not differ from fiber in chemotherapy-related GI symptoms

GI: gastrointestinal, RCT: randomized clinical trial, EPA: eicosapentaenoic acid, DHA: docosahexaenoic acid, PUFAs: polyunsaturated fatty acids, QoL: quality of life.

## 4. Discussion

Cancer-associated cachexia remains a prevalent and critical health issue, further complicating the morbidity, quality of life, and survival of cancer patients. Cancers of the GI tract are more susceptible to cachexia development due to a higher risk of interruption in essential functions of nutrient absorption and digestion than other cancer types. In addition, cancer patients with cachexia lose primarily lean or fat-free body mass over fat mass, which contributes to metabolic dysregulation, given that protein is a less efficient source for energy production but is more readily available [73,74,75]. The recently developed Cancer Cachexia Risk Score includes many of the above risk factors, namely cancer site, cancer stage, time from symptom onset to hospitalization, appetite loss, body mass index, skeletal muscle index, and neutrophil-lymphocyte ratio [76]. The risk score was determined prior to abdominal surgery in over 16,000 patients with GI cancers and showed a high sensitivity of 75.1% in correctly predicting cancer cachexia development post-surgery. In addition to the Cancer Cachexia Risk Score [76], the Geriatric Nutritional Risk Index (GNRI) may benefit from early detection of nutritional risk by utilizing the clinically available data on weight and albumin level [77].

These risk score assessments can assist clinicians in making early decisions about nutritional and pharmacological support for GI cancer patients who are likely to develop cancer cachexia. However, current pharmacological treatments are limited in reducing the progression or preventing the development of cancer cachexia. Patients and providers have tried various supplements to prevent or treat cancer cachexia, with limited evidence supporting their benefit. Nutritional supplements such as fish oil, melatonin, probiotics, and green tea, at least, are usually not associated with adverse effects if administered as part of the diet and a nutrition plan. Findings suggest these supplements play a role in reducing systemic inflammation, restoring gut microbiome balance, and promoting digestion, which leads to improving quality of life or slowing the progression of cachexia.

A significant gap in our understanding of the beneficial effects of dietary supplements in cancer and cancer-associated cachexia remains the translational aspect from cell lines and animals to humans. Several factors contribute to the sustained lack of understanding and evidence-based clinical approach to the use of supplements. One such factor remains the inconsistent composition of supplements via chemical analysis. It complicates and often makes it impossible to compare one product with another, even if it contains the same active compound on the label. Secondly, no clear guidance on dosages and duration of supplement use may cause insignificant findings in the studies. Another factor is the late diagnosis of cancer cachexia and delayed adequate nutritional support that would allow for the preventive use of such supplements in pre-cachectic patients. While we have a good understanding of the effectiveness of chemotherapy based on the cancer stage, further research is warranted for the appropriate time points to initiate nutritional supplements to prevent or reverse cancer cachexia. One study indicated that the fish oil group showed significantly longer survival than a placebo group within the same mGPS of one or two categories [32]. It suggested monitoring blood biomarkers of the CRP and albumin levels are critical to have beneficial effects of fish oil, where the mGPS score is assigned to 0, 1, or 2 based on the CPR and albumin levels, with ‘0′ indicating no abnormal CRP or albumin [78].

Our review has strengths. We identified the current evidence of the effectiveness of supplement use and the gaps in supplements used for GI cancer-associated cachexia through a narrative review. Additionally, aside from the discussed supplements, several other approaches appear promising in preventing and managing cancer cachexia in GI cancers, most notably cannabinoids for their appetite-stimulant, anti-nausea, and immune-modulating properties [79,80].

There are limitations to this review. First, many studies in this area had small sample sizes, various dosages of the supplements, heterogeneous intervention periods, and diverse composition of the supplements. It may have led to non-significant findings in many studies. Second, we did not find any meta-analysis studies, which would benefit concluding outcomes when studies with large sample sizes were not available. Thus, we only included RCTs and other prospective studies. Finally, our findings can be generalizable to the supplements mentioned above.

## 5. Conclusions

While more clinical trial evidence is warranted, healthcare providers and patients with GI cancer at risk of developing or having been diagnosed with cancer cachexia may consider integrating dietary supplements as part of a nutritional plan to reduce treatment-associated adverse effects and improve the quality of life. The evidence demonstrated the support for using fish oil, melatonin, and specific probiotics. The clinical benefit of green tea preparations is limited; however, potential benefits need further evaluation, since they are commonly available to community members.

## Figures and Tables

**Figure 1 nutrients-15-03391-f001:**
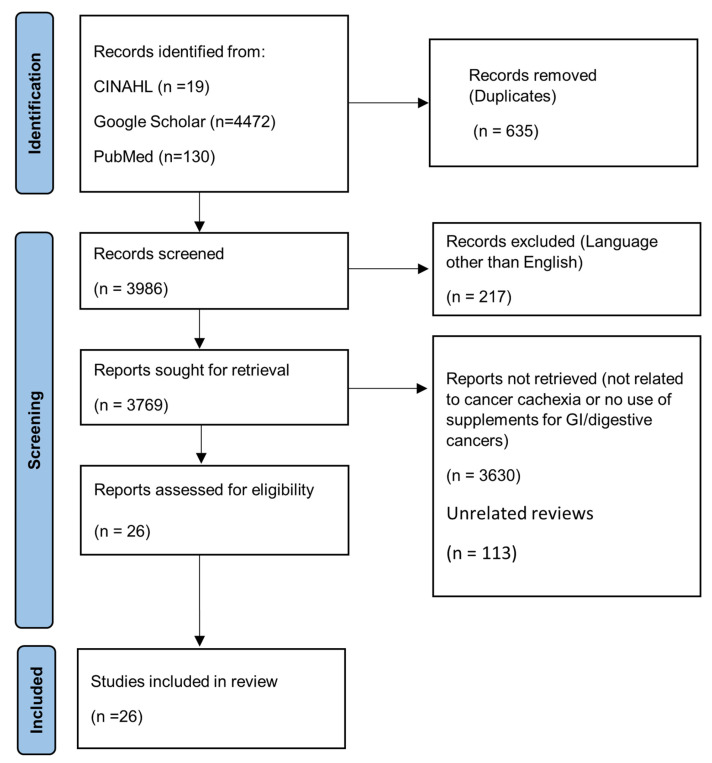
PRISMA flowchart for article inclusion [27].

**Table 1 nutrients-15-03391-t001:** Search results before screening and inclusion of articles.

Search Term	PubMed	Google Scholar	CINAHL
“gastrointestinal cancer” AND “cachexia” AND			
“fish oil”	34	1090	1
“melatonin”	6	443	3
“probiotics”	10	313	2
“green tea”	2	181	0
“supplements”	61	2320	10
“digestive cancer” AND “cachexia” AND			
“fish oil”	6	22	1
“melatonin”	1	8	1
“probiotics”	0	10	0
“green tea”	0	6	0
“supplements”	10	79	1

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
