# Peer review of "Relevance of Dietary Supplement Use in Gastrointestinal-Cancer-Associated Cachexia"

_nutrients, 2023, doi:10.3390/nu15153391_

Round 1

Reviewer 1 Report (Previous Reviewer 2)

In the revised manuscript, the authors have increased the searching area of the topic, while the number of articles involved in the text did not change. Moreover, as the main point I concerned before, the authors just listed the results of the papers, without any summaries and perspectives, even though they added a table at the end of result section. Therefore, I am sorry to say that the revised version do not fulfill my requirements.

Minor editing of English language required

Author Response

We have now included summary statements for each supplement at the end of the respective section. The summary is meant to address the lack of perspectives that the reviewer would like to see incorporated. This summary is then further supported by the discussion section. We intentionally kept the perspective in the discussion section so to not confuse the reader by providing interpretation or discussion in the results section.

We also added more information and detail to each study included in the table based on prior comments.

Reviewer 2 Report (New Reviewer)

Comments to the article:

1) The last sentence of the ‘Introduction’ section should be supported by references from literature.

2) In the ‘Methodology’, sufficient criteria were not provided for the eligibility of works for review (inclusion and exclusion criteria, e.g. only randomised studies, on humans, on animal models, in English, etc.)

3) In the ‘Results’ section, the authors indicate that they generally relied on interventional or observational studies, which may limit the scientific value of the review. Please comment.

4) The limitations of the prepared review should be indicated.

The linguistic quality of the text, apart from minor errors, is sufficient.

Author Response

1) The last sentence of the ‘Introduction’ section should be supported by references from literature.

Response: we revised the sentence to emphasize the focus of the review.

2) In the ‘Methodology’, sufficient criteria were not provided for the eligibility of works for review (inclusion and exclusion criteria, e.g. only randomised studies, on humans, on animal models, in English, etc.)

Response: we included the inclusion/exclusion criteria in the methods section with a new subheading 2.2. Furthermore we also expanded on the search strategy section by including that we followed the PRISMA guidelines.

3) In the ‘Results’ section, the authors indicate that they generally relied on interventional or observational studies, which may limit the scientific value of the review. Please comment.

Response: we did indeed primarily rely on clinical trials to determine the effectiveness of the discussed supplements. Although preclinical results were discussed as they contributed to the understanding of how a supplement may provide benefits, the focus was on clinical outcomes. This does consider both interventional as well as observational studies.

4) The limitations of the prepared review should be indicated.

Response: we now added a limitations section to the discussion. It is clear that observational and interventional studies on the included supplements are heterogenous, lack a standardized product, and usually are small in sample size.

Round 2

Reviewer 1 Report (Previous Reviewer 2)

In the revised version of the mauscript, the authors have covered all my concerns.

Reviewer 2 Report (New Reviewer)

The authors have covered my comments.

This manuscript is a resubmission of an earlier submission. The following is a list of the peer review reports and author responses from that submission.

Round 1

Reviewer 1 Report

This is a review article on the use of nutritional supplements for gastrointestinal cancer. the manuscript is well described and updated, but it strikes me as strange that none of the authors work in the field of nutrition.

In the abstract section, authors provides results from animal studies but the aim of the study was to evaluate the effects os supplementation in cachexia cancer-patients. You can use animal studies to discuss the mechanisms of action but not in the abstract. And the results need to suggest the benefits but not provide statement of the supplements evaluated in the review.

In the introduction section, it was not addressed how cachexia in cancer modifies the nutritional status of patients, and how this impairs the response to clinical treatment, which justifies the use of supplements to alleviate the symptoms of cachexia. Also, in lines 59-64 it is understood that cancer cachexia and malnutrition are separate things and, in fact, they happen together. Cachexia is a severe state of malnutrition and this concept needs to be improved.

In line 90 there is no consensus on low cost supplements, especially in developing countries with high tax rates on dietary supplements.

In the method section it is not attached to the search strategy, which makes it difficult to reproduce the results. In addition, only the result of 26 articles used for this review does not seem appropriate to me to conduct a review article.

In general, the results for each supplement should be divided into before and after surgery or clinical treatments (radiotherapy or chemotherapy) to facilitate understanding of the action of these supplements. Another important fact is that the dose and the type of fatty acids of the supplements in the studies cited was not mentioned in all studies. "Omega-3 fatty acids" represents a lot of different fatty acids profile. In the case of fish oil, which fatty acid profile did it have? Rich in linolenic or EPA and DHA? For how long were  the patients supplemented? Without this information it is difficult to understand the results. The same works fot melatoni, green tea and probiotics, the results should separate before and after surgery or clinical treatments, and doses and time of supplementation.

There is no conclusion section, authors should provide the conclusion of this review.

Reviewer 2 Report

In the review, Saunjoo L. Yoon and Oliver Grundmann attempted to summary the evidences between dietary supplement use and gastrointestinal cancer-associated cachexia. Such issue has been addressed in several previous reviews. Therefore, the novelty of the manuscript should be considered as questioned. Moreover, the scope of the review should be reconsidered, due to only a few of papers involved. In the result section, the authors just listed the results of the papers, without any summaries and perspectives, which further reduced the value the review.  

Minor editing of English language required.

Reviewer 3 Report

This is a systematic review without meta analysis. It provides useful information on complementary interventions for an important problem such as neoplastic cachexia. It is a simple but well-conducted review.